# Knowledge and Expectations of Orthodontic Retention Among Individuals Seeking Orthodontic Treatment in Saudi Arabia: A Cross-Sectional Study

**DOI:** 10.3390/dj14010021

**Published:** 2026-01-04

**Authors:** Narmin M. Helal, Nujud O. Saber, Mohammed F. Almalki, Osama A. Basri

**Affiliations:** 1Department of Pediatric Dentistry, Faculty of Dentistry, King Abdulaziz University, Jeddah 21589, Saudi Arabia; nhilal@kau.edu.sa; 2Department of General Dentistry, Faculty of Dentistry, King Abdulaziz University, Jeddah 21589, Saudi Arabia; malmalki0578@stu.kau.edu.sa; 3Dentistry Department, King Faisal Hospital and Research Center, Jeddah 21499, Saudi Arabia; obasri@gmail.com

**Keywords:** orthodontic retention, orthodontic retainers, treatment outcome, Saudi Arabia

## Abstract

**Background/Objectives:** Orthodontic retention is a critical phase of treatment aimed at maintaining teeth in corrected positions and preventing relapse, which may occur in up to 70% of cases. Successful retention depends on both orthodontists’ application of effective strategies and patient compliance in retainer use and maintenance. While previous studies have focused on orthodontists’ retention protocols, less is known about patients’ knowledge and expectations. This study assessed knowledge and expectations of orthodontic retention among individuals seeking orthodontic treatment in Saudi Arabia. **Methods:** A cross-sectional survey was conducted at the orthodontic clinics of King Abdulaziz University, Faculty of Dentistry, Jeddah, Saudi Arabia. Patients on the waiting list who had not yet begun treatment were invited to participate; guardians completed surveys for minors. Data were collected using a validated, culturally adapted questionnaire and analyzed with descriptive statistics and multivariable logistic regression. **Results:** Of 243 patients contacted, 161 responded (66.3%). While 62.1% were aware of retention appliances, only 31.1% believed they were always required. Overall, 57.1% expected retention to last 1–3 years and were divided between bonded and removable retainers. Orthodontists were the most trusted information source, while social media was least trusted. Family history and higher education were associated with greater awareness and support for lifelong retention, though most associations were not significant after multiple-testing correction. **Conclusions:** Despite awareness of retention appliances, misconceptions persist. Family history, education, and age influenced perceptions, underscoring the need for tailored education and guidance toward reliable orthodontic information sources.

## 1. Introduction

Orthodontic retention is the stage of treatment in which active tooth movements are completed, and the focus shifts to maintaining the teeth in their newly aligned position [1]. Orthodontic retention is a critical component of comprehensive orthodontic care, as relapse, which refers to the tendency for teeth to return to their original maligned positions, has been reported in up to 70% of cases, even after standardized orthodontic treatment [2,3,4]. To prevent relapse, various types of retainers are commonly used, including fixed retainers, removable Hawley retainers, and vacuum-formed retainers [5].

Recent technological developments in orthodontic retention focus on remote monitoring, innovative retainer materials, and digital workflows. Remote monitoring using thermal sensors, mobile apps, and AI-driven platforms such as Dental Monitoring allows clinicians to track patient compliance and retainer fit more efficiently. Advances in materials, including nickel-titanium and cobalt-chrome, along with digitally fabricated retainers via CAD-CAM and 3D printing, aim to improve retainer durability, precision, and customization, potentially enhancing long-term treatment stability [6,7].

Recent technological advances in orthodontics have introduced tools to monitor and support patient compliance during the retention phase. Sensor-based devices embedded in retainers objectively track wear, revealing that adherence is often suboptimal, while mobile reminder applications have shown limited impact on retainer use, stability, or periodontal outcomes [8]. The integration of tele-orthodontics, sensor-based retainers, and reminder systems offers promising avenues to support patient compliance and oral hygiene, although the role of social media as a communication tool between clinicians and patients remains unclear [9].

The success of the retention phase depends on both the orthodontist and the patient. Orthodontists must have a thorough understanding of the etiology of relapse and the appropriate application of retention strategies. Simultaneously, patient compliance in wearing and maintaining retainers as instructed by their orthodontist is essential for achieving long-term stability [10].

Despite the widespread use of orthodontic retainers, patients’ understanding, and perceptions of post-treatment retention remain limited. Prior research showed that among patients at least two years post-treatment, awareness of post-treatment tooth movement was limited, with only 74% of vacuum-formed retainer users and 47.1% of bonded retainer users recognizing changes [11]. Prior research has thoroughly investigated orthodontists’ attitudes and preferences toward different retention protocols [12,13,14,15,16]. A growing body of literature has also begun to address patients’ perception of orthodontic retention and their role in maintaining treatment outcomes [17,18,19,20]. Previous studies have highlighted the importance of incorporating patients’ preferences and values into treatment planning, as this can enhance satisfaction, adherence, and overall treatment effectiveness [21,22]. Therefore, understanding how patients perceive orthodontic retention is essential for orthodontists to tailor their communication and education strategies to meet patients’ needs, which will ultimately improve long-term treatment outcomes.

In Saudi Arabia, a previous study conducted by Al-Jewair et al. [13] discussed the orthodontic retention protocols followed by orthodontists in Saudi Arabia. However, patient perspectives in this context remain unexplored. Thus, this study aimed to address this knowledge gap by evaluating the knowledge of orthodontic retention among individuals seeking orthodontic treatment in Jeddah, Saudi Arabia.

## 2. Materials and Methods

### 2.1. Study Design

This cross-sectional observational survey-based study was conducted after obtaining approval from the Ethical Research Committee of the King Abdulaziz University Faculty of Dentistry (KAUFD), Jeddah, Saudi Arabia (Protocol Code: 106-06-25). The study adheres to the STROBE (Strengthening the Reporting of Observational Studies in Epidemiology) guidelines for reporting cross-sectional research [23]. A completed STROBE checklist is included in Appendix A: A completed STROBE checklist.

### 2.2. Study Setting and Participants

Participants were recruited from the orthodontic clinic patients waiting list at KAUFD. Patients who had not yet started orthodontic treatment were identified through a review of clinic records and invited to participate via text messages sent to their registered phone numbers. Prior to accessing the questionnaire, all participants were presented with a digital consent form outlining the purpose and nature of the study. Participation was voluntary, and no personally identifiable information was collected. Individuals who consented to participate received a secure electronic survey, created using the Google Forms platform. For participants under 18 years of age, surveys were completed by their parents or legal guardians. A reminder message was sent two weeks after the initial invitation to encourage participation for the ones who did not reply to the initial invitation.

### 2.3. Sampling and Inclusion Criteria

A convenience sampling strategy was used. The survey was distributed to all eligible patients listed on the orthodontic treatment waiting list at KAUFD between January 2024 and June 2025. Eligibility was determined based on the following criteria: adults (≥18 years) currently seeking orthodontic treatment, as well as parents or legal guardians of minors (<18 years) seeking treatment on behalf of their children, were included. Individuals who were currently undergoing or had previously completed orthodontic treatment were excluded. All invited individuals who provided informed consent were included.

### 2.4. Data Collection Tool

A validated questionnaire, adapted from a previous study [18], was used to collect data from eligible participants. The questionnaire was originally developed in English. To ensure clarity and cultural relevance, it was translated into Arabic using forward and backward translation methods by a certified translation agency. Arabic responses were re-translated into English by the research team for validation and analysis. Face validity was confirmed by two orthodontists. Content validity was assessed using the Item Content Validity Index (I-CVI), based on four domains: relevance, clarity, simplicity, and ambiguity. Items were rated on a 4-point scale [24]. With three experts, excellent content validity was achieved with the Scale-Content Validity Index (S-CVI) = 1.0, indicating full agreement [25].

The questionnaire was then pilot tested with 20 laypeople that have not undergone orthodontic treatment. Feedback from this pilot test was used to refine the questionnaire, and the responses from the pilot phase were excluded from the main analysis.

In addition to the items from the validated questionnaire, one Appendix A: The complete survey question was added to explore participants’ most trusted source of information regarding orthodontic retention. The responses to this item were analyzed separately and not considered in the assessment of knowledge or expectation. Questionnaire adaptation and validation process is shown in Figure 1.

The questionnaire consisted of three main sections. The first section gathered demographic information, including age, gender, and prior orthodontic treatment experience. The second section assessed participants’ knowledge of orthodontic retention, including awareness of retention protocols and types of retainers, as well as their expectations regarding retainers, including anticipated duration of use, compliance, and related concerns. The third section explored participants’ perceived confidence in various sources of information related to orthodontic retention. This section included a Appendix A: The complete survey question that asked participants to rate their trust in different sources, including orthodontists, general dentists, friends and family, online search engines (e.g., Google), social media platforms (e.g., Instagram and TikTok), and Artificial Intelligence (AI) chatbots (e.g., ChatGPT). Each source was rated using a five-point Likert scale ranging from “Very confident” to “Very not confident.” The questionnaire was available in both Arabic and English. All questions were set as required items to avoid incomplete answers. The complete survey is included in Appendix A The complete survey question.

### 2.5. Statistical Analysis

Analyses were conducted using STATA statistical software, version 17 (StataCorp, College Station, TX, USA). Descriptive analysis was conducted to assess participants’ sociodemographic and their knowledge and expectation on orthodontic retention, as well as participants’ confidence levels in various sources of information on orthodontic retention. All factors were summarized into frequencies and percentages. Adjusted binary logistic regression models were constructed, after checking for assumption violations, to examine the relationship between participants’ knowledge, expectation, and confidence (dependent variables) and their social-demographic characteristics (independent variables). In the regression models, each independent variable was paired with each dependent variable, and the results were reported in odds ratios as well as the corresponding 95% confidence intervals. Also, *p*-values at a significance level of 5% were adjusted for multiple testing within each family of outcomes using the Benjamini–Hochberg false discovery rate procedure. Additionally, the adjusted q-values were reported.

## 3. Results

### 3.1. Socio-Demographic Characteristics

Among the 243 patients contacted, 161 completed the survey, yielding a response rate of 66.3%. As shown in Table 1, the majority were aged 18–24 years (37.3%) and 52.2% were female. Regarding education level, 39.8% held a bachelor’s degree. Over half (55.3%) were seeking treatment for a child under 18 years, and 57.8% reported a family history of orthodontic treatment. Most participants (71.4%) were self-motivated to seek orthodontic treatment.

### 3.2. Knowledge on Orthodontic Retention

As summarized in Table 2, 62.1% of participants were aware that retention appliances are used after orthodontic treatment. 31.1% believed these appliances are required in all cases. Regarding treatment outcomes, 80.8% thought that a perfect result guarantees long-term stability, while 56.5% acknowledged that teeth can shift even without appliances.

### 3.3. Expectations on Orthodontic Retention

As shown in Table 3, most participants (57.1%) expected retention to last one and three years, with 13.7% anticipating lifelong use. Most participants considered achieving a stable treatment result to be extremely important (59.6%). Recall visits were expected every three (41.6%) or six months (32.9%), with responsibility for stability mainly placed on patients/parents (77%) and orthodontists (70.2%). Finally, 93.8% considered charging for recall visits inappropriate.

### 3.4. Confidence Levels in Various Sources of Information on Orthodontic Retention

As shown in Table 4, orthodontists were the most trusted source, with 88.2% very confident. Friends/family and online searches elicited mostly “somewhat confident” responses (41.6% and 42.2%, respectively). Social media was least trusted (7.5% very confident, 33.5% somewhat confident), while AI chatbots had 14.9% very confident and 37.3% somewhat confident respondents.

### 3.5. Relationship of Participants’ Socio-Demographic Characteristics and Their Knowledge and Expectation Toward Orthodontic Retention

In Table 5, adjusted binary logistic regression showed that participants with close family members who had orthodontic experience were nearly five times more likely to know about retention devices (aOR = 4.91, *p* < 0.001) and three times more likely to believe that perfect treatment guarantees stability (aOR = 3.05, *p* = 0.011). Participants aged 35 and older were 75% less likely to think the retention phase should be lifelong (aOR = 0.25, *p* = 0.017). Those with tertiary education had 7.83 times more likely to think retention phases were lifelong (aOR = 7.83, *p* = 0.002, q = 0.076). Self-seeking participants were more aware that teeth can move without appliances (aOR = 2.41, *p* = 0.021) and less likely to support charges for recall visits (aOR = 0.21, *p* = 0.029). Notably, these significant associations, except for the family history and awareness of continuous orthodontic device use were at the nominal level (*p* < 0.05) but were not significant after correction for multiple testing (q > 0.05).

### 3.6. Relationship Between Participants’ Confidence in Information Sources with Their Demographic Characteristics

Table 6 shows that participants with tertiary education were less confident in friends/family (aOR = 0.45, *p* = 0.025), self-motivated individuals were more confident in online searches (aOR = 2.74, *p* = 0.011), and those with a family history of orthodontic treatment trusted AI chatbots more (aOR = 2.77, *p* = 0.003). However, none of these associations remained significant after false discovery rate adjustment (q > 0.05) and should be considered exploratory trends.

## 4. Discussion

This cross-sectional study assessed the knowledge and expectations of individuals seeking orthodontic treatment in Saudi Arabia regarding orthodontic retention. Although over half of participants were aware that appliances are used for retention after orthodontic treatment, notable misconceptions were still observed. Most participants expected the retention phase to last one to three years and considered achieving stable results highly important. Orthodontists and general dentists were considered the most reliable sources of information, whereas social media was viewed with greater skepticism.

This study revealed that 62.1% of participants were aware that appliances are used for retention after orthodontic treatment, higher than the 46.3% reported in the previous study conducted in Switzerland [18]. Regarding the necessity of retention appliances, 31.1% of our participants believed they are required in all cases, which is comparable to the 32.6% observed in the prior study [18]. In terms of treatment outcomes, 80.8% of participants in this study believed that achieving a perfect treatment result guarantees long-term stability, compared to 52.8% in the aforementioned study. Additionally, 56.5% recognized that teeth could move even without orthodontic appliances, a proportion lower than the 77.8% reported previously [18]. These differences may be explained by methodological variations between the studies, including data collection approaches and cultural context.

Supporting these findings, a cross-sectional study conducted at Ziauddin University, Karachi [26], found that 64.9% of patients believed teeth could not move without orthodontic appliances, while 56.1% believed parents and guardians were responsible for maintaining treatment results, and 42.9% considered dentists/orthodontists accountable for treatment stability. These comparisons suggest that awareness of retention appliances may be increasing, potentially due to broader access to orthodontic care, family exposure, and digital information. Nonetheless, persistent gaps remain in understanding long-term stability and treatment expectations, emphasizing that basic awareness does not necessarily translate into accurate understanding. Taken together, these findings highlight the need for comprehensive, culturally sensitive educational strategies that address both the practical aspects of retention and the biological processes influencing treatment stability.

Participants’ confidence in orthodontic information varied considerably according to the source, highlighting a clear hierarchy of trust. Professional sources were considered the most reliable, with 88.2% of participants expressing high confidence in orthodontists and 44.1% in general dentists. In contrast, social media platforms were viewed as the least trustworthy, with only 7.5% reporting high confidence. AI sources, including chatbots, occupied an intermediate position, with 14.9% of participants expressing high confidence. These findings are consistent with recent evaluations of online orthodontic content. A content analysis of TikTok videos on orthodontic retention [27], found that only 22.1% of videos were classified as high-quality, despite orthodontists contributing 37.8% of the content. Similarly, an evaluation of YouTube videos uploaded by dental professionals [28] reported that although orthodontists accounted for the majority of content, the overall quality of information was deficient, particularly in domains such as the need for regular retainer reviews. Furthermore, a recent study by Martina et al. [29] showed that although chatbots can provide helpful orthodontic information, they may give inaccurate guidance on controversial topics, highlighting the need for professional oversight. Studies evaluating AI-generated responses, including ChatGPT on clear aligners [30] and AI chatbots in orthognathic surgery [31], similarly show that while accuracy can be moderate to high, readability and accessibility may limit utility for patients without advanced education. Additionally, it underscores the need for dedicated research evaluating AI responses specifically in the context of orthodontic retention. Together, these results highlight the essential role of professional guidance. As patients increasingly rely on social media and AI tools, orthodontists should proactively provide accurate, evidence-based, patient-centered content to correct misconceptions and promote adherence.

Family history, education, age, and consultation pathway influenced participants’ knowledge, expectations, and confidence regarding orthodontic retention. Having close family members with prior orthodontic treatment was the strongest predictor of awareness of retention appliances and was also linked to the belief that perfect treatment guarantees stability, suggesting that familial exposure can simultaneously reinforce accurate knowledge and perpetuate misconceptions. This aligns with a cross-sectional study conducted at Ziauddin University, Karachi, which found that participants with a positive family history of orthodontic treatment had higher odds of being aware that retainers are used after treatment (OR: 0.366; 95% CI: 0.198–0.677; *p* = 0.001) [26]. Participants who self-sought consultation demonstrated greater confidence in online health information, reflecting proactive health-seeking behavior. This aligns with findings from a study examining digital health literacy and web-based health information seeking behaviors among the Saudi population, which indicated that higher digital health literacy correlates with increased confidence in navigating online health information [32]. However, increased engagement with digital sources may also expose patients to inaccurate information.

This study has several strengths. It included all eligible patients on the orthodontic treatment waiting list during the study period, minimizing selection bias and providing a comprehensive snapshot of the target population. By assessing patients before treatment initiation, this study offers a perspective that is underrepresented in the literature, highlighting misconceptions and expectations that have not yet been influenced by clinical experience. This contributes new evidence to the limited research on pre-treatment understanding of orthodontic retention, particularly within Middle Eastern populations. The use of validated survey items and structured data collection also ensured consistency and reliability of the responses. However, certain limitations should be considered. The sample was obtained using a convenience sampling approach at a single institution, which may limit the generalizability of the findings to broader populations. Additionally, no a priori power calculation was performed, as the study included all available and eligible patients during the data collection period. The cross-sectional design prevents causal interpretations, and responses may be subject to self-reporting bias. While a range of sociodemographic factors was analyzed, some associations were only nominally significant after correction for multiple testing. Future research may benefit from multicenter sampling to improve representativeness, as well as longitudinal designs to track how knowledge and expectations evolve throughout treatment. Interventional studies evaluating tailored educational strategies, such as structured counseling, visual aids, or digital resource guidance, could further clarify the most effective methods to enhance patient understanding, retention compliance, and long-term treatment outcomes. Additionally, developing and implementing clinician-friendly checklists or guiding questions to assess patients’ knowledge and perceptions of orthodontic treatment and the post-treatment retention phase could help identify misconceptions and allow for more personalized education, ultimately promoting realistic expectations and improved adherence.

## 5. Conclusions

This study shows that, while many patients seeking orthodontic treatment in Saudi Arabia are aware of retention appliances, key misconceptions persist regarding their necessity, duration, and the potential for relapse. Most participants expected retention to last only one to three years, and only a minority believed retainers are required in all cases. Confidence in information sources varied greatly, with orthodontists ranked highest and social media the lowest. Family history, education, and age influenced knowledge and expectations, although most associations were not significant after multiple-testing correction. These findings highlight the need for early, structured patient education that clearly explains the purpose of retention, the likelihood of long-term relapse, and appropriate retainer use. Providing accurate guidance and directing patients toward reliable sources may help reduce misconceptions and support better long-term treatment outcomes.

## Figures and Tables

**Figure 1 dentistry-14-00021-f001:**
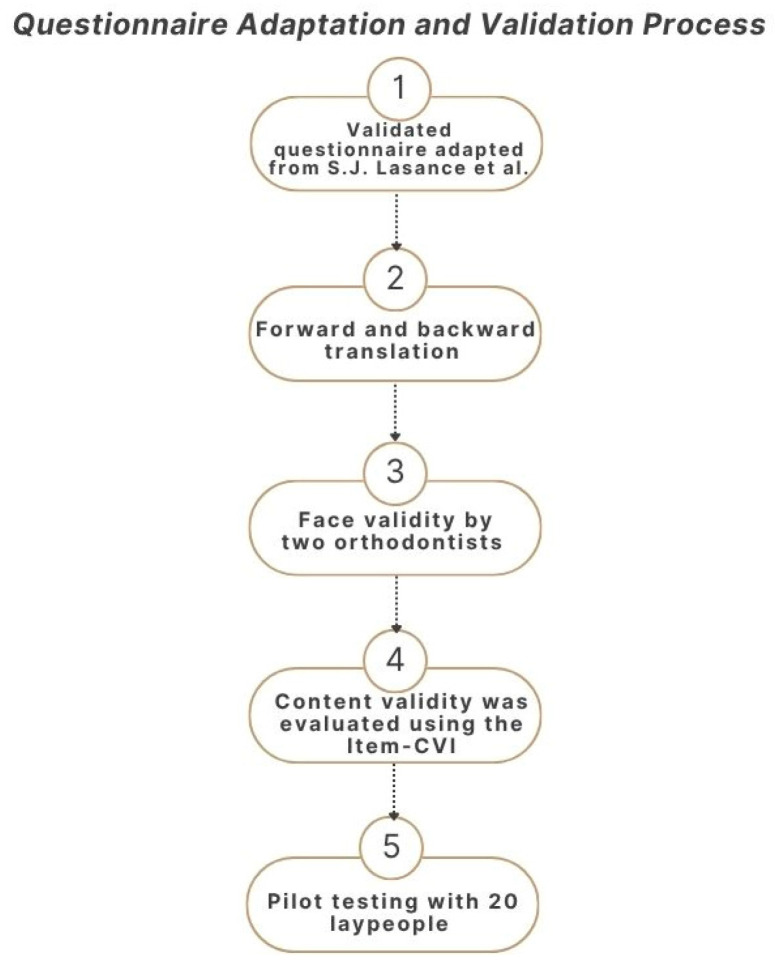
Questionnaire Adaptation and Validation Process [18].

**Table 1 dentistry-14-00021-t001:** Socio-demographic characteristics of participants (n = 161).

	n	%
**Age:**		
18–24	60	37.3
25–34	27	16.8
35–44	25	15.5
45–54	34	21.1
55+	15	9.3
**Gender:**		
Male	77	47.8
Female	84	52.2
**Educational level:**		
Uneducated	13	8.1
High school or lower education	63	39.1
Bachelor’s degree	64	39.8
Postgraduate degree	21	13.0
**Who is seeking treatment?**		
Respondent	72	44.7
Respondent’s child under 18	89	55.3
**Previous experiences with orthodontics within close family:**		
No	68	42.2
Yes	93	57.8
**Reason for consultation:**		
Referral by someone	46	28.6
Self-motivated	115	71.4

**Table 2 dentistry-14-00021-t002:** Participants’ level of knowledge on orthodontic retention (*n* = 161).

	n	%
**Are you aware that appliances are used for retention after orthodontic treatment?**		
No	61	37.9
Yes	100	62.1
**How often do you think such appliances are necessary?**		
In rare cases	9	5.6
In some cases	73	45.3
In most cases	29	18.0
In all cases	50	31.1
**In which cases do you consider retention necessary?** (*N_a_* = 216) *		
After comprehensive orthodontic treatment	61	37.9
After orthodontic treatment during growth	50	31.1
After orthodontic treatment in adults	22	13.7
After treatment with extractions	8	5.0
In all cases	75	46.6
**Do you believe a perfect treatment result can guarantee stability?**		
No	31	19.3
Yes	130	80.8
**Do you think that teeth can also move without orthodontic appliances?**		
No	70	43.5
Yes	91	56.5

*N_a_*, number of participants that answered the question; *, Multiple choice question, so the summary % adds up to more than 100%.

**Table 3 dentistry-14-00021-t003:** Participants’ expectation regarding orthodontic retention (*n* = 161).

	n	%
**How long do you think the retention phase should be?**		
Less than a year	36	22.4
1–3 years	92	57.1
3–10 years	11	6.8
Lifelong	22	13.7
**How important is a stable result for you?**		
Not important	2	1.2
Ambivalent	7	4.4
Rather important	56	34.8
Extremely important	96	59.6
**Which type of retention device would you favor?**		
Bonded device	81	50.31
Removable device	80	49.69
**At which interval do you believe recall visits are necessary?**		
Every 3 months	67	41.61
Every 6 months	53	32.92
Yearly	30	18.63
Every 2nd year	7	4.35
Every 5th year	4	2.48
**Who do you consider responsible for the stability after orthodontic treatment?** (*N* = 263) *		
General dentist	26	16.1
orthodontist	113	70.2
Patient/or parent	124	77.0
**Do you think it is appropriate to charge for recall visits?**		
No	151	93.79
Yes	10	6.21

*, Multiple choice question, so the summary % adds up to more than 100%.

**Table 4 dentistry-14-00021-t004:** Participants’ confidence in information related to orthodontics retention received from different sources (*n* = 161).

	Very Not Confident (%)	Somewhat Not Confident (%)	Neutral (%)	Somewhat Confident (%)	Very Confident (%)
**Orthodontist**	1	0	1	17	142
	(0.6)	(0.0)	(0.6)	(10.6)	(88.2)
**General dentist**	0	3	19	68	71
	(0.0)	(1.9)	(11.8)	(42.2)	(44.1)
**Friends and family**	3	23	38	67	30
	(1.9)	(14.3)	(23.6)	(41.6)	(18.6)
**Online searches**	9	24	44	68	16
	(5.6)	(14.9)	(27.3)	(42.2)	(9.9)
**Social media**	12	34	49	54	12
	(7.5)	(21.1)	(30.4)	(33.5)	(7.5)
**Artificial Intelligence Chatbots**	13	26	38	60	24
	(8.1)	(16.2)	(23.6)	(37.3)	(14.9)

**Table 5 dentistry-14-00021-t005:** Statistically significant predictors of participants’ beliefs and attitudes toward orthodontic retention.

	Gender ^1^	Age ^2^	Education ^3^	Past Experiences ^4^	Consultation Reason ^5^
**Participant is aware that retention devices are being used after orthodontic treatment**	NS	NS	NS	aOR = 4.91 CI = (2.35–10.24) *p*-value < 0.001 q-value = 0.001	NS
**Participant believes a perfect result guarantees stability**	NS	NS	NS	aOR = 3.05 CI = (1.30–7.16) *p*-value = 0.011 q-value = 0.321	NS
**Participant thinks that the retention phase should be lifelong**	NS	aOR = 0.25 CI = (0.08–0.8) *p*-value = 0.017 q-value = 0.608	aOR = 7.83 CI = (2.11–29.1) *p*-value = 0.002 q-value = 0.076	NS	NS
**Participant thinks teeth can move without orthodontics**	NS	NS	NS	NS	aOR = 2.41 CI = (1.14–5.10) *p*-value = 0.021 q-value = 0.630
**Patient favors removable over fixed retainer**	NS	NS	NS	NS	NS
**Participant believes it is appropriate to charge for recall visits**	NS	NS	NS	NS	aOR = 0.21 CI = (0.05–0.85) *p*-value = 0.029 q-value = 0.885

aOR, adjusted Odd Ratio; NS, not significant; CI, Confident Interval. Significance level: *p* < 0.05 (nominal); q < 0.05 (FDR-adjusted); ^1^ Female (reference); ^2^ 18–34 years (reference) compared to older; ^3^ Uneducated/high schoolers (reference) compared to bachelors and postgraduates; ^4^ Family member did not undergo treatment (reference) compared to family member had treatment; ^5^ Referred by someone (reference) compared to self-sought consultation.

**Table 6 dentistry-14-00021-t006:** Participants’ confidence in information sources.

	Gender ^1^	Age ^2^	Education ^3^	Past Experiences ^4^	Consultation Reason ^5^
**Orthodontist**	NS	NS	NS	NS	NS
**General dentist**	NS	NS	NS	NS	NS
**Friends and family**	NS	NS	aOR = 0.45 CI = 0.23–0.90 *p*-value = 0.025 q-value = 0.836	NS	NS
**Online searches**	NS	NS	NS	NS	aOR = 2.74 CI = 1.27–5.94 *p*-value = 0.011 q-value = 0.360
**Social media**	NS	NS	NS	NS	NS
**Artificial Intelligence Chatbots**	NS	NS	NS	aOR = 2.77 CI = 1.40–5.50 *p*-value = 0.003 q-value = 0.118	NS

aOR, adjusted Odd Ratio; NS, not significant; CI, Confident Interval. Significance level: *p* < 0.05 (nominal); q < 0.05 (FDR-adjusted); ^1^ Female (reference); ^2^ 18–34 years (reference) compared to older; ^3^ Uneducated/high schoolers (reference) compared to bachelors and postgraduates; ^4^ Family member did not undergo treatment (reference) compared to family member had treatment; ^5^ Referred by someone (reference) compared to self-sought consultation.

## Data Availability

The data supporting the findings of this study are available from the corresponding author upon reasonable request.

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
