# Peer review of "Knowledge and Expectations of Orthodontic Retention Among Individuals Seeking Orthodontic Treatment in Saudi Arabia: A Cross-Sectional Study"

_dentistry, 2026, doi:10.3390/dj14010021_

Round 1
Reviewer 1 Report
Comments and Suggestions for Authors
Dear Authors, Thank you for the opportunity to review your interesting manuscript. The topic contributes to a better understanding of patients’ awareness and expectations regarding orthodontic retention in Saudi Arabia. However, several aspects of the paper require clarification and refinement to improve its clarity, rigor, and overall impact.
Please consider the following suggestions:
Title:
Revise the title to make it more concise and fluent. For example:
Knowledge and Expectations of Orthodontic Retention Among Patients in Saudi Arabia: A Cross-Sectional Study.
Study Power:
Please include information about the study’s statistical power.
Sampling Method:
The use of a convenience sampling approach should be explicitly acknowledged as a limitation in the Limitations section, as it may affect the generalizability of the findings.
Results Section:
Clarify the Results section to improve readability and remove redundant information.
Discussion:
- Divide the Discussion into clear subsections for better organization (e.g., “Comparison with Previous Studies,” “Clinical Implications,” “Limitations and Future Directions”).
- Emphasize the novel aspects and contributions of your study to the existing literature.
- Add missing references where appropriate (e.g., line 266).
- Remove repetitive content.
Conclusions:
Rewrite the Conclusions to make them less general and to highlight the new insights and practical implications your study provides.
Author Response
We sincerely thank the reviewer for their insightful and constructive comments.
|
Comment 1: Title: Revise the title to make it more concise and fluent. For example: Knowledge and Expectations of Orthodontic Retention Among Patients in Saudi Arabia: A Cross-Sectional Study. |
|
Response 1: We thank the reviewer for this helpful suggestion. The title has been revised for clarity and conciseness as follows: “Patients’ Knowledge and Expectations Regarding Orthodontic Retention in Saudi Arabia: A Cross-Sectional Study.” |
|
Comments 2: Study Power: Please include information about the study’s statistical power. Sampling Method: The use of a convenience sampling approach should be explicitly acknowledged as a limitation in the Limitations section, as it may affect the generalizability of the findings. |
| Response 2:
Thank you for this valuable comment. We have now clarified the sampling approach and acknowledged the lack of an a priori power calculation in the Limitations section. As well as the use of convenience sampling approach. The following sentence was added in page 11 line 324-328: “However, certain limitations should be considered. The sample was obtained using a convenience sampling approach at a single institution, which may limit the generalizability of the findings to broader populations. Additionally, no a priori power calculation was performed, as the study included all available and eligible patients during the data collection period.” |
|
Comments 3: Results Section: Clarify the Results section to improve readability and remove redundant information. Response 3: We have revised all Results subsections (3.1–3.6) to be more concise and focused, while retaining all key information. Tables have been retained for detailed data. Comment 4: “Divide the Discussion into clear subsections for better organization (e.g., ‘Comparison with Previous Studies,’ ‘Clinical Implications,’ ‘Limitations and Future Directions’).” Response 4: Thank you for this helpful suggestion. The Discussion has now been restructured into subsections to enhance clarity and readability. |
|
Comment 5: “Emphasize the novel aspects and contributions of your study to the existing literature.” Response 5: We appreciate this comment. We have strengthened the Discussion by highlighting the novelty of assessing pre-treatment patients, which provides insight into misconceptions prior to clinical influence. We also emphasized that the study contributes to the limited evidence available in Middle Eastern populations regarding orthodontic retention expectations. The following sentence was added in page 11 line 319-322: "By assessing patients before treatment initiation, this study offers a perspective that is underrepresented in the literature, highlighting misconceptions and expectations that have not yet been influenced by clinical experience. This contributes new evidence to the limited research on pre-treatment understanding of orthodontic retention, particularly within Middle Eastern populations." Comment 6: “Add missing references where appropriate (e.g., line 266).” Response 6: The missing reference has been added at the indicated location. |
|
Comment 7: “Remove repetitive content.” Response 7: Repetitive statements regarding awareness levels and patient education needs have been consolidated to avoid redundancy and improve flow. The revised Discussion is more concise while preserving all essential interpretations of the findings. |
|
Comments 8: Rewrite the Conclusions to be less general and highlight new insights and practical implications. new conclusion: " This study provides new insights into patients’ understanding and expectations of orthodontic retention in Saudi Arabia. While more than half of participants recognized the role of retention appliances, many held misconceptions about long-term stability. Family history, education, age, and consultation pathway were key factors influencing knowledge, expectations, and confidence in information sources. These findings underscore the need for orthodontists to provide structured, early counseling that goes beyond simply describing retainer types. Clear discussions about long-term stability, the likelihood of relapse, and expected maintenance should be integrated into the initial treatment planning visit. Additionally, as many patients rely on informal or digital sources of information, with varying levels of confidence, guiding them toward reliable educational materials may help improve retention adherence. In practice, targeted patient education and personalized communication strategies may reduce misconceptions, increase compliance, and ultimately contribute to better long-term treatment outcomes." |
We believe that these revisions enrich the manuscript and we sincerely appreciate your constructive feedback, which has significantly strengthened the clarity and focus of our manuscript.
Reviewer 2 Report
Comments and Suggestions for Authors Dear authors, congratulations on the topic; it is highly relevant and important. The introduction must be more focused on the existing problem with the perception of retention. The sample size calculation is missing. The reference list is relatively short, and only 10 out of 26 references were published within the last five years.Author Response
We sincerely thank the reviewer for their insightful and constructive comments.
Reviewer comment 1:
"The introduction must be more focused on the existing problem with the perception of retention."
Response 1:
We have revised the Introduction to focus more clearly on the existing problem of patients’ perceptions of orthodontic retention.
Reviewer comment 2:
"The sample size calculation is missing."
Response 2:
Regarding sample size, no a priori calculation was performed because we included all available and eligible participants during the study period. Additionally, we have clarified the sampling approach and acknowledged the lack of an a priori power calculation in the Limitations section. The following sentence has been added to the manuscript in page 11, line 324-328:
"However, certain limitations should be considered. The sample was obtained using a convenience sampling approach at a single institution, which may limit the generalizability of the findings to broader populations. Additionally, no a priori power calculation was performed, as the study included all available and eligible patients during the data collection period."
Reviewer comment 3:
"The reference list is relatively short, and only 10 out of 26 references were published within the last five years."
Response:
The reference list has been updated with more recent publications to ensure current and relevant citations.
We appreciate your time and expertise in reviewing our work, and we believe the revisions have improved both the clarity and impact of our study.
Reviewer 3 Report
Comments and Suggestions for Authors
Dear authors,
your cross-sectional investigation addresses an important and under-explored topic in Saudi orthodontic literature: patients' knowledge and expectations regarding the orthodontic retention phase. The work is characterized by a well structured methodology, an appropriate statistical analyses and an excellent clarity of presentation.
Here are a few tips to improve your already excellent work.
- The Results section repeats almost verbatim the data already reported in the tables. I suggest summarizing the text and reserving the analysis of the most relevant trends for the Discussion section;
- Among the studies analyzing the reliability and accuracy of AI-chatbots in providing information on orthodontics, I would mention the recent work by Martina et al. (doi: 10.3390/dj13080343), which compares different chatbots as educational sources on orthodontic topics.
Author Response
First, We would like to express our appreciation for your constructive and insightful feedback.
Reviewer Comment:
“The Results section repeats almost verbatim the data already reported in the tables. I suggest summarizing the text and reserving the analysis of the most relevant trends for the Discussion section.”
Response:
We thank the reviewer for this valuable observation. In response, we have carefully revised the Results section to provide a more concise narrative that avoids redundancy with the tables. The revised text now highlights only key findings, while detailed numerical values remain within the tables. Additionally, interpretation of trends and comparisons with the literature have been moved to the Discussion section to maintain clarity and avoid overlap.
Reviewer Comment:
“Among the studies analyzing the reliability and accuracy of AI-chatbots in providing information on orthodontics, I would mention the recent work by Martina et al. (doi: 10.3390/dj13080343), which compares different chatbots as educational sources on orthodontic topics.”
Response:
We appreciate this helpful recommendation. The study by Martina et al. has now been cited and discussed in the revised Discussion section, particularly within the subsection addressing the reliability of AI-generated orthodontic information. The citation strengthens our argument regarding variability in chatbot accuracy and further supports the need for professional guidance when patients seek digital orthodontic information.
Lastly, Your insightful comments have helped us improve the depth and relevance of our study, and we are grateful for your contribution to enhancing the quality of this study.
Reviewer 4 Report
Comments and Suggestions for Authors
This study stands out as a unique contribution to understanding patients knowledge and perceptions of orthodontic retention.
To further strengthen the manuscript, I suggest including a brief discussion on recent advances in orthodontic retention appliances, particularly newer options that extend beyond traditional lingual wire retainers. Highlighting developments such as digitally fabricated retainers and enhanced clear thermoplastic materials would provide readers with a more updated view of current clinical practice.
It may also be useful to mention the emerging technologies for monitoring patient compliance, such as mobile reminder applications or smart retainers with tracking features, which can support patient cooperation and improve overall treatment outcomes.
Finally, to enhance the practical applicability of your findings, it would be beneficial to include a checklist or set of guiding questions that orthodontists can use to assess patients knowledge and perceptions about orthodontic treatment and the post treatment retention phase. Such tools could help clinicians identify misconceptions and deliver more personalized education and guidance toward reliable digital resources, ultimately promoting more realistic expectations.
Minor corrections: In table 6 , You mentioned “Antiracial Intelligence Chatbots.” Did you mean to say “Artificial Intelligence Chatbots”
Author Response
We sincerely thank the reviewer for their insightful and constructive comments.
Reviewer Comment:
“To further strengthen the manuscript, I suggest including a brief discussion on recent advances in orthodontic retention appliances, particularly newer options that extend beyond traditional lingual wire retainers. Highlighting developments such as digitally fabricated retainers and enhanced clear thermoplastic materials would provide readers with a more updated view of current clinical practice.”
Response:
We thank the reviewer for this suggestion. We have included a paragraph in the introduction in page 2, line 49-63, on recent developments in orthodontic retention, highlighting digitally fabricated retainers, clear thermoplastic materials, and novel options such as CAD-CAM nickel-titanium retainers. These additions provide readers with an updated view of contemporary clinical practice and emerging alternatives to traditional lingual wire retainers.
Reviewer Comment:
“It may also be useful to mention the emerging technologies for monitoring patient compliance, such as mobile reminder applications or smart retainers with tracking features, which can support patient cooperation and improve overall treatment outcomes.”
Response:
We appreciate the reviewer’s recommendation. We have incorporated a discussion of emerging technologies to monitor patient adherence, including sensor-based retainers and mobile reminder applications in the introduction in page 2, line 69-73.
Reviewer Comment:
“Finally, to enhance the practical applicability of your findings, it would be beneficial to include a checklist or set of guiding questions that orthodontists can use to assess patients’ knowledge and perceptions about orthodontic treatment and the post-treatment retention phase. Such tools could help clinicians identify misconceptions and deliver more personalized education and guidance toward reliable digital resources, ultimately promoting more realistic expectations.”
Response:
We have addressed this by adding a recommendation in the Future Directions section to develop clinician-friendly checklists or guiding questions in the discussion, in page 11, line 336-340.
Reviewer Comment:
“In Table 6, you mentioned ‘Antiracial Intelligence Chatbots.’ Did you mean to say ‘Artificial Intelligence Chatbots’?”
Response:
Thank you for pointing this out. This was a typographical error, and it has been corrected to “Artificial Intelligence Chatbots” in Table 6.
We believe that these revisions enrich the manuscript by integrating recent technological advancements and practical strategies, enhancing both its scientific and clinical relevance.